# Clinical Occurrences in the Neurorehabilitation of Dogs with Severe Spinal Cord Injury

**DOI:** 10.3390/ani13071164

**Published:** 2023-03-25

**Authors:** Débora Gouveia, Sara Fonseca, Carla Carvalho, Ana Cardoso, António Almeida, Óscar Gamboa, Rute Canejo-Teixeira, António Ferreira, Ângela Martins

**Affiliations:** 1Arrábida Veterinary Hospital, Arrábida Animal Rehabilitation Center, 2925-538 Setubal, Portugal; 2Superior School of Health, Protection and Animal Welfare, Polytechnic Institute of Lusophony, 1950-396 Lisboa, Portugal; 3Faculty of Veterinary Medicine, Lusófona University, 1749-024 Lisboa, Portugal; 4Faculty of Veterinary Medicine, University of Lisbon, 1300-477 Lisboa, Portugal

**Keywords:** functional neurorehabilitation, dogs, clinical occurrences, plegia, deep pain, SCI

## Abstract

**Simple Summary:**

Patients with a severe spinal cord injury are highly predisposed to clinical occurrences such as pressure ulcers and urinary tract infections. This study aimed to evaluate the frequency of clinical occurrences in dogs with neurological conditions submitted to intensive inpatient neurorehabilitation, in order to develop preventive measures. From a total of 488 dogs, 79.5% had at least one clinical occurrence—the most frequent being neurogenic bladder (58%), followed by diarrhea (35.5%), urinary incontinence (21.3%), and fecal incontinence (20.5%). Preventive measures such as point-of-care ultrasound monitoring, positioning strategies, kinesiotherapy exercises, and early implementation of neurorehabilitation can help prevent some of the clinical occurrences frequently described in the literature.

**Abstract:**

This prospective observational clinical study in a population of tetraplegic and paraplegic dogs (*n* = 488) with or without deep pain sensation, similar to humans ASIA A and B, investigated the prevalence of clinical occurrences in a rehabilitation center with a hospitalization regime between 15 days and 9 months. A checklist of occurrences was used for easy identification and monitoring, resulting in a total of 79.5% occurrences. There were 58% of dogs with neurogenic bladder, 35.5% with diarrhea, 21.3% with urinary incontinence, and 20.5% with fecal incontinence. A low incidence of respiratory problems (e.g., pneumonia) and urinary tract infections may suggest the efficacy of some applied measures in this study, such as thoracic and abdominal POCUS evaluation, positioning strategies, physical exercises, respiratory kinesiotherapy, and early implementation of a functional neurorehabilitation protocol. These can be essential measures to prevent clinical occurrences, mainly in breeds such as the French Bulldog and the Dachshund.

## 1. Introduction

Several studies in human medicine have reported a higher risk and incidence of clinical occurrences in patients with a severe spinal cord injury (SCI) [1,2]. The most common concomitant clinical events described in paraplegic or tetraplegic patients are the presence of pressure sores, urinary tract infections (UTIs), intestinal disorders, respiratory complications [1,2,3,4], and muscle weakness and contractures [5], which play an important role in mobility [6].

In human medicine, pressure sores can result in serious complications. With high morbidity rates for patients with SCI, classified as ASIA A or B (American Spinal Injury Association) needing a specific approach to the severity of the situation [6,7].

The second most common cause is UTIs, due to permanent or temporary micturition disorders, which increase the risk of urinary infections that can lead to a life-threatening situation for these patients [2,8].

On the contrary, dysfunction of the gastrointestinal system has a great impact on quality of life, especially in human patients. Thus, decreased colon mobility, bowel obstruction syndrome, constipation, and fecal incontinence are major concerns [2]. These patients may have a high risk of fecal incontinence due to reduced/absent perineal sensitivity and voluntary control of the external anal sphincter [9].

Respiratory complications are related to the severity of the injury. Therefore, cervical SCI can result in pulmonary dysfunction, such as pneumonia and atelectasis, most commonly in the first five days after injury [2,4]. In human medicine, respiratory occurrences have a prevalence of 36–83%, of which 80% occur in patients with cervical lesions [4].

The literature review showed an extremely high development in human medicine on this subject, but few papers have been published in the veterinary literature, although there is a complete description of palliative care in patients with neurological conditions [10].

Supportive care recommendations in these patients are of primary importance, resulting in the need to work with specialized nursing teams [11]—especially in dogs classified as grade 0 or 1, according to the Modified Frankel Scale (MFS) [12], and OFS 0 and OFS 1, according to the Open Field Score (OFS) [13]. These recommendations may include bathing, harness support, positioning to prevent pressure sores, use of elastic bands, and bladder and fecal incontinence management [14,15].

Furthermore, there are some musculoskeletal changes related to disuse and secondary atrophy, which can include bones, cartilage, muscles, ligaments, and tendons. For example, 3–11 weeks of immobilization may decrease the stifle joint flexion, with a possible 9–50% reduction in cartilage thickness [16]. Moreover, in these patients, the most vulnerable muscles to disuse atrophy are the postural ones that comprise a relatively large proportion of type I (slow-twitch) muscle fibers [16]. In addition, many of these SCI dogs have concomitant osteoarticular problems [17].

Thus, it is essential to obtain early information about the possibility of clinical occurrences in an intensive neurorehabilitation center, in order to develop and implement preventive measures that may increase the quality of life of dogs with neurological conditions.

The present prospective observational study aimed to illuminate the prevalence of clinical occurrences in dogs undergoing hospitalized neurorehabilitation in a referral rehabilitation center. As a secondary aim, this study intended to develop preventive measures for the main clinical occurrences identified.

The study hypothesis was to consider the existence of a positive relationship between the possibility of clinical occurrences in patients with neurological conditions undergoing neurorehabilitation in a hospitalization setting.

## 2. Material and Methods

This prospective observational study was conducted between 1 September 2016 and 1 February 2023 at the Arrábida Veterinary Hospital (Arrábida Animal Rehabilitation Center, Setúbal, Portugal/Lisbon Animal Rehabilitation and Regeneration Center, Lisbon, Portugal), after approval by the Lisbon Veterinary Medicine Faculty Ethics Committee (N/Refª 001/209), and after receiving the owner’s consent.

The population study (*n* = 488) was based on dogs with neurological conditions that remained in the rehabilitation center for more than two weeks and up to nine months. The inclusion criteria selected all SCI dogs with plegic presentation, with or without deep pain (DP) perception, meaning grade 0 or 1 (according to the MFS) and OFS 0 or OFS 1. With regard to etiology, dogs were selected with compressive myelopathy, including intervertebral disk herniation (IVDH) (early and late chondroid metaplasia) and lumbosacral stenosis; and with non-compressive myelopathy, such as acute non-compressive nucleus pulposus extrusion (ANNPE), hydrated nucleus pulposus extrusion (HNPE), and fibrocartilaginous embolic myelopathy (FCEM). Additionally, trauma dogs with a more compressive or contusion injury were included and divided according to the magnetic resonance imaging (MRI) or computed tomography (CT) results.

Sample characterization (*n* = 488), according to age, weight, sex, breed, neuroanatomical localization, etiology, DP perception, and clinical occurrences, is described in Table 1.

### 2.1. Study Design

At admission, all 488 dogs were submitted to a functional neurorehabilitation (FNR) consultation performed by a certified canine rehabilitation professional (CCRP, Tennessee University) examiner/instructor. The examination took place in a controlled environment room and all data were recorded (Canon EOS Rebel T6 1300 D camera, Canon, Taichung City, Taiwan). This included a systematic and strict evaluation considering mental status, gait, proprioception and postural reactions, vertebral palpation (spinal hyperesthesia assessment), hindlimb peripheral reflexes (patellar, cranial tibial, withdrawal, and crossed-extensor reflexes), forelimb peripheral reflexes (carpus radial extensor, withdrawal, and crossed-extensor reflexes), perineal and cutaneous trunci reflexes, muscle tone evaluation, and pain perception.

With regard to pain perception, special care was taken during the evaluation of the DP in the medial and lateral digits on each of the hind- and fore-limbs. Pain was also assessed in the perineal region and the tip and base of the tail.

A check list was created to record all clinical occurrences at admission and throughout the 15 days to 9 months of the study (Table 2).

At the end of each neurorehabilitation appointment, all dogs underwent an FNR protocol. These were performed on a similar basis for all participants, with some adaptations according to the specificities of each patient, such as etiology, neuroanatomical location, and presence or absence of DP.

The overall FNR protocol was based on different neurorehabilitation methods and modalities, which are described in Figure 1.

### 2.2. Data Collection

Data were collected from all 488 dogs, including the continuous quantitative variables, age (<7 years or ≥7 years) and weight (<15 kg or ≥15 kg). The categorical nominal variables collected were sex, breed (pure-breed or mixed-breed), chondrodystrophy, neuroanatomical localization (cervical, thoracolumbar, or lumbosacral region), etiology (compressive or non-compressive myelopathy), DP perception (present or absent), sternal recumbency (present or absent), and clinical occurrences (present or absent). The categorical binomial variables of occurrences considered for the present study were neurogenic bladder, urinary incontinence, urinary infection, fecal incontinence, diarrhea, vomiting, dermatitis, pressure sores, kennel cough, aspiration pneumonia, bronchopneumonia, muscle atrophy, spasticity, conjunctivitis, episcleritis, corneal ulcers, pyrexia, pancreatitis, discospondylitis, and progressive myelomalacia.

### 2.3. Statistical Analysis

Data records for all dogs were documented using Microsoft Office Excel 365^®^ (Microsoft Corporation, Redmond, WA, USA), and processed in IBM SPSS Statistics 25^®^ software (International Business Machines Corporation, Armonk, NY, USA). Kolmogorov–Smirnov normality test (for *n* > 50), arithmetic mean, minimum, maximum, standard deviation (SD), and standard error of the mean (SEM) were performed for the continuous variables of age and weight. Descriptive statistics with frequency analysis were carried out for all categorical nominal variables. Chi-square tests were also performed to verify the relevant level of significance proven by a *p*-value of ≤0.05.

## 3. Results

In this prospective observational clinical study, from the 488 dogs with plegic presentation, there were 59.8% (292/488) males and 40.2% (196/488) females. Descriptive analysis of the continuous variables, age and weight, are reported in Table 3, demonstrating a mean age of 5.57 years old and a mean weight of 13.57 kg. The Kolmogorov–Smirnov normality test (*n* > 50) was performed for the continuous variables of age and weight, and revealed an absence of normality (*p* ≤ 0.001) with regard to the study population. 

For the binominal variable “breed,” it was observed that 23% (112/488) were mixed-breed dogs and 77% (376/488) were pure-breed dogs, which included the French Bulldog (*n* = 99) as the most prevalent pure-breed, followed by the Dachshund (*n* = 46), Yorkshire Terrier (*n* = 34), Labrador Retriever (*n* = 25), Pekinese (*n* = 22), Poodle (*n* = 16), Beagle (*n* = 14), Jack Russell Terrier (*n* = 10), Dobermann (*n* = 9), Chihuahua (*n* = 7), Portuguese Podengo (*n* = 7), Shih-tzu (*n* = 7), Pinscher (*n* = 6), Dalmatian (*n* = 5), Grand Danois (*n* = 5), Spitz (*n* = 5), English Bulldog (*n* = 4), Pug (*n* = 4), Basset Hound (*n* = 3), Cocker Spaniel (*n* = 3), German Shepherd (*n* = 3), Greyhound (*n* = 3), Maltese Dog (*n* = 3), Portuguese Water Dog (*n* = 3), Weimaraner (*n* = 3), American Staffordshire Terrier (*n* = 2), Bernese Mountain Dog (*n* = 2), Bullmastiff (*n* = 2), English Setter (*n* = 2), Epagneul Breton (*n* = 2), Golden Retriever (*n* = 2), Rottweiler (*n* = 2), Bordeaux Mastiff (*n* = 1), Bouvier Bernois (*n* = 1), Boxer (*n* = 1), Briard (*n* = 1), Cane Corso (*n* = 1), Fox Terrier (*n* = 1), German Shorthaired Pointer (*n* = 1), Giant Poodle (*n* = 1), Pitbull (*n* = 1), Rafeiro Alentejano (*n* = 1), Rhodesian Ridgeback (*n* = 1), Schnauzer (*n* = 1), Siberian Husky (*n* = 1), Soft Coated Wheaten Terrier (*n* = 1), West Highland White Terrier (*n* = 1), and Whippet (*n* = 1). Thus, it was observed that 55.1% (269/488) were chondrodystrophic and 44.9% (219/488) were non-chondrodystrophic breeds. Additionally, 21.9% (107/488) of the dogs were brachiocephalic. 

With regard to etiology, 93% (454/488) of the dogs presented with compressive myelopathy (IVDH; lumbosacral stenosis syndrome), and 7% (34/488) with non-compressive myelopathy (ANNPE; HNPE; FCEM). As for the neuroanatomical localization, it verified a prevalence of 72.7% (355/488) in the thoracolumbar region, 26% (127/488) in the cervical region, and 1.2% (6/488) in the lumbosacral region. Thus, at admission, 74% (361/488) of the dogs presented as paraplegic and 26% (127/488) as tetraplegic. 

Considering the neurorehabilitation examination, 72.5% (354/488) were DP-positive (DPP), while 27.5% (134/488) were DP-negative (DPN). From all 488 dogs, 54.9% (268/488) were not able to present sternal recumbency, while 45.1% (220/488) showed the ability to maintain sternal recumbency.

As for the occurrences (Table 4), at least one clinical occurrence was observed in 79.5% (388/488) of the dogs included in the study, while 20.5% (100/488) presented none. The most prevalent occurrences were neurogenic bladder with 58% (283/488), and diarrhea with 35.5% (173/488).

Regarding the presence of occurrences and the categorical variable “etiology,” no significance was found. However, as for the neuroanatomical localization and the presence of occurrences (*n* = 388), a significant association (*χ*^2^(2, *n* = 488) = 150.703, *p* ≤ 0.001) was observed, with 85% (330/388) in dogs with thoracolumbar localization, 13.7% (53/388) in cervical localization, and 1.3% (5/388) with lumbosacral localization.

Furthermore, in the present study, a significant association between the presence of occurrences and the neurological condition of paraplegia or tetraplegia (*χ*^2^(1, *n* = 488) = 150.368, *p* ≤ 0.001) was verified, and from a total of 361 paraplegic dogs, 335 had shown occurrences.

In addition, the relationship between the binominal categorical variable “DP” and the prevalence of these occurrences proved to be significant (*χ*^2^(1, *n* = 488) = 40.926, *p* ≤ 0.001), with only two of the 134 DPN dogs having no occurrences.

When comparing brachycephalic breeds and bronchopneumonia (*n* = 107), only one brachycephalic dog with thoracolumbar localization showed bronchopneumonia. The same was observed for the presence of aspiration pneumonia (*n* = 9), which did not show significance with this breed, with only one French Bulldog with a cervical lesion. Moreover, from the 26% (127/488) of patients with cervical lesions, six developed aspiration pneumonia. The presence of this clinical occurrence with regard to the ability to maintain sternal recumbency showed a significant association (*χ*^2^(1, *n* = 488) = 4.274, *p* = 0.039), and 88.9% (8/9) of aspiration pneumonias were related to the absence of sternal recumbency ability.

A significant association between brachycephalic breeds and the presence of dermatitis (*χ*^2^(1, *n* = 488) = 220.325, *p* ≤ 0.001) was also confirmed, with 74.1% (80/108) of brachycephalic dogs having dermatitis, of which 77 were French Bulldogs. Likewise, the clinical occurrence of vomiting (*n* = 71) was significant (*χ*^2^(1, *n* = 488) = 8.566, *p* = 0.023) in the same breeds, with a prevalence of 35.2% (25/71).

Ophthalmic occurrences, presented in 24.4% (119/488) of the study population, showed strong significance (*χ*^2^(1, *n* = 488) = 168.246, *p* ≤ 0.001) in the brachycephalic breeds (*n* = 107), with 44% (47/107) of the dogs having conjunctivitis, 33.6% (36/107) having episcleritis, and 9.3% (9/107) having corneal ulcers (Figure 2). No significance was detected when considering ophthalmic occurrences and the ability to maintain sternal recumbency.

As for thoracolumbar localization and the binominal variables, neurogenic bladder and diarrhea, both were significant (*χ*^2^(2, *n* = 488) = 235.631, *p* ≤ 0.001 and *χ*^2^(2, *n* = 488) = 84.608, *p* ≤ 0.001, respectively). Additionally, considering only thoracolumbar localization (*n* = 355), 44.5% (158/355) of the dogs showed both occurrences. Regarding cervical neuroanatomical localization and the occurrences referred to above, no significance was observed, since no dogs presented both clinical signs.

Concerning fecal incontinence (*n* = 100), all dogs were DPN and had thoracolumbar (*n* = 98) or lumbosacral (*n* = 2) localization (*χ*^2^(2, *n* = 488) = 44.366, *p* ≤ 0.001). The same was observed for urinary incontinence (*n* = 104), with 102 dogs having thoracolumbar localization, of which 101 were DPN, demonstrated by strong significance (*χ*^2^(1, *n* = 355) = 240.108, *p* ≤ 0.001). Additionally, when considering the occurrences of UTI and DP, significance was found (*χ*^2^(1, *n* = 488) = 44.300, *p* ≤ 0.001), with 24 UTI occurrences in DPN dogs.

Lastly, no significance was noticed between the presence of pressure sores and muscle atrophy, with neither paraplegia nor tetraplegia, although in 63 paraplegic dogs with pressure sores, 65% (41/63) were DPN, shown by the strong significance of (*χ*^2^(1, *n* = 361) = 25.562, *p* ≤ 0.001). When comparing the occurrence of pressure sores (*n* = 95) and the ability to maintain sternal recumbency, strong significance was also observed (*χ*^2^(1, *n* = 488) = 25.154, *p* ≤ 0.001), with 77.9% (74/95) of the dogs with pressure sores unable to maintain sternal recumbency.

## 4. Discussion

In the present study, the population was equivalent in terms of gender distribution, although with a greater representation of males [18]. The mean age was 5.57 years (9.05 variance), with 77% (376/488) being pure-breed dogs, of which 99 were French Bulldogs and 46 were Dachshunds, thus representing the most prevalent breeds. These data agree with the results of Dickinson and Bannasch (2020) [19], who reported the presence of compressive SCI secondary to chondroid metaplasia with high expression of the fibroblast growth factor 4 retrogene on chromosome 12 (CFA12 FGF4 retrogene) in both breeds, although at different ages, but within the average age of our study [20].

As for weight, an average of 13.57 kg was observed, but with a high variance of 111.08 kg, due to the great diversity of breeds (*n* = 376), agreeing with the non-normal distribution in the Kolmogorov–Smirnov test, and highlighting deviation related to the higher prevalence of the two breeds mentioned above [18]. 

Considering spinal cord compressive myelopathies as the etiology with 93% (454/488), the higher prevalence of IVDH is understandable, once again due to the chondroid metaplasia typical in chondrodystrophic breeds [21]. Such breeds were the most common in our study, with 55.1% (269/488), emphasizing the expression of the CFA12 FGF4 retrogene as the greatest risk factor for extrusive compressive metaplasias of the intervertebral disc [21,22,23,24].

The 7% (34/488) non-compressive myelopathy may be due to the extremely acute character of ANNPE and FCEM, resulting in contusion of the medullary parenchyma with the release of local inflammatory mediators (e.g., cytokines and excitatory neurotransmitters), which cause a greater probability of tissue ischemia [17,25,26], also justifying possible plegia of one to four limbs depending on the neurolocation of the SCI [17,27].

All of these are compatible with our population of plegic dogs, with a total of 74% (361/488) paraplegic, and 26% (127/488) tetraplegic dogs, reflecting a sample that can benefit from FNR. This is also in agreement with different authors, who refer to the suggestion that intensive neurorehabilitation protocols are not necessary for all patients with incomplete spinal cord injuries, supported by moderate-level evidence [28].

With regard to neuroanatomical localization, the highest prevalence was in the thoracolumbar region, with 72.7% (355/488), which is the most prevalent site for extrusion compressive myelopathies [29,30,31]. The strong significance obtained regarding localization and the presence of occurrences (*χ*^2^(2, *N* = 488) = 150.703, *p* ≤ 0.001) was also demonstrated by the 85% (330/388) of dogs who showed thoracolumbar neurolocation, contrasting with the 13.7% (53/388) showing cervical neurolocation, of the total amount of dogs with occurrences (*n* = 388)—although, with a bias associated with the heterogeneous distribution of neurolocation.

In this population of plegic dogs, 27.5% (134/488) were DPN, considered an important population to investigate. A study by Martins et al. (2021) [32] showed that from 94 DPN dogs subjected to intensive neurorehabilitation protocols in a hospital, 35 recovered DP and 22 developed spinal reflex locomotion. It was also reported that 74.3% of the DP recovery occurred after 30 days of hospitalization and that the peak in which the dogs achieved spinal reflex locomotion was after 60 days of hospitalization, justifying the need for prolonged hospitalization in some patients, which in our study was from 15 days to 9 months. Likewise, studies with plegic patients with signs of residual spinal shock have reported the need for at least 15 days to reach ambulation of 68% [17,32,33,34,35]. Considering the hospitalization regimen for chronic patients with compressive myelopathy, DPP dogs took an average of 47 days to achieve ambulation, and 78% of the DPN dogs achieved spinal reflex locomotion in 2.5 months [36], confirming the need for FNR through hospitalization, never for a period of less than 15 days.

Regarding clinical occurrences, there was an incidence of 79.5% (388/488), suggesting a high percentage of cases in these dogs in a clinical setting of hospitalization for FNR.

The presence of neurogenic bladder was highly prevalent, with 58% (283/488), which may be explained by 72.7% of the dogs having T3–L3 neurolocation [15,37,38], confirmed by the significance in the present study (*χ*^2^(2, *N* = 488) = 235.631, *p* ≤ 0.001). Thus, secondary bladder distension with urinary retention, and possible dyssynergy between the detrusor muscle and the ureteral sphincter, may occur in these patients, followed by the inhibition of sphincter relaxation with the contraction of the detrusor muscle, justifying high residual urinary volumes [39].

A further 21.3% (104/488) of dogs had urinary incontinence, possibly explained by the presence of DPN in 27.5% of dogs and lumbosacral etiology in 1.2% (6/488) of dogs. Atonic bladders can occur with lower motor neuron injuries [40], or secondary-to-severe SCI due to poor urinary management or involuntary contraction of the detrusor muscle [15].

Urinary management may be complex in patients with neurological conditions, with a high risk of developing UTIs due to incomplete emptying [40,41] and bacteriuria [34,38,40]. Additionally, urinary catheterization as a prolonged procedure predisposes to UTIs, stimulating a harmful pathogenic cellular environment. In human medicine, the presence of proteases has been studied, as well as their possible interaction with pathogenic agents that may have ascending tropism, leading to pyelonephritis, bacteremia, and septicemia [42,43,44,45]. Therefore, there is positive evidence that manual expression should be taught to the rehabilitation team in order to avoid urinary catheterization [28]. This manual expression technique was performed in the present study by the neurorehabilitation team (Figure 3), as a reference in the field, possibly justifying the low UTI percentage of 6.1% (30/488). In specific cases of urinary catheterization, a closed system should be used with a sterile environment, and a urinary culture must be done after catheter removal [15].

In the comparison between the presence of occurrences and DP perception, a high significance (*χ*^2^(1, *N* = 488) = 40.926, *p* ≤ 0.001) was observed, with only 2 of the 134 DPN dogs having no occurrences. This is probably because DPN dogs need more days of hospitalization, thus having a higher predisposition, in addition to the physiological changes inherent to plegic presentation [14]. There was also a strong significance among DPN patients of thoracolumbar neurolocation and the presence of urinary incontinence (*χ*^2^(1, *N* = 355) = 240.108, *p* ≤ 0.001). As for the clinical sign of diarrhea, its prevalence was 35.5% (173/488), also demonstrating strong significance with thoracolumbar neurolocation (*χ*^2^(2, *N* = 488) = 84.608, *p* ≤ 0.001). In human medicine, a decrease in vagal afferent receptor sensitivity to neuropeptides and neurotransmitters has been demonstrated, resulting in a blockage of descending motor bundles and their communication with the lumbosacral segmental circuit, followed by a pathophysiological regulation of the intrinsic neural circuit of the colon [46].

Dogs with fecal incontinence are reported after thoracolumbar injuries, even when ambulation is recovered, which was demonstrated in this study by a strong significance between thoracolumbar injuries, fecal incontinence, and DPN patients (*χ*^2^(2, *N* = 488) = 44.366, *p* ≤ 0.001). The pathophysiological mechanisms are not well known, but may be based on the epidural hematoma associated with IVDH, resulting in a crucial problem that could lead to patient euthanasia [47].

In this study, 44.5% (158/355) of dogs with thoracolumbar neurolocation presented both clinical signs of neurogenic bladder and diarrhea. In these types of patients, there is evidence of a link between fecal incontinence, dermatological lesions, and contamination of the vulva with secondary UTIs [15]. The suggestion of a gastrointestinal or hypoallergenic diet, in addition to local trichotomy, strict asepsis with chlorhexidine 0.05%, and the application of zinc ointments, as well as the use of diapers [14,45], may be important preventive measures. A similar relationship was not verified in this study due to the residual prevalence of UTIs. Thus, continuous bladder distention and colon irritation could be due to an excessive dysregulation of the sympathetic preganglionic neuron depolarization [46], whose cells are located in lamina VII of the spinal cord gray matter [48,49].

Furthermore, the clinical sign of vomiting was significant when related to brachycephalic breeds (*χ*^2^(1, *N* = 488) = 8.566, *p* = 0.023), which is corroborated by the high predisposition of gastrointestinal diseases in these breeds [50], given its specific features—such as elongation of the soft palate, macroglossia, stenotic nostrils, and laryngeal disease [51,52], which are all present in brachycephalic syndrome. Poncet et al. (2005) [53] reported that 97.3% of dogs with observed brachycephalic signs have gastroduodenal problems diagnosed by fibroendoscopy, presenting the clinical sign of vomiting, which has been referred to by different authors [51,52,54], as well as in the present study.

The same breeds had a significant association with the frequency of ophthalmic problems (*χ*^2^(1, *N* = 488) = 168.246, *p* ≤ 0,001), a statement already reported by previous researchers [55,56]. Thus, as preventive measures when clinical signs were detected, an Elizabethan collar was applied to prevent self-trauma, followed by artificial drops to lubricate, or topical serum every 4–6 h and, if necessary, a topical nonsteroidal anti-inflammatory to control pain [57,58].

Likewise, brachycephalic dogs presented a high predisposition for dermatitis, as verified by this study, with high significance of (*χ*^2^(1, *N* = 488) = 220.325, *p* ≤ 0.001) and a total prevalence of 74.1% (80/108) of brachycephalic dogs with dermatitis, of which 77 were French Bulldogs. This is in agreement with previous studies on healthy brachycephalic dogs that showed that French Bulldog are the most common breed experiencing dermatological problems [59,60,61].

Regarding the prevalence of pressure sores, 19.5% (95/488) of dogs demonstrated their occurrence. Of these, 63 dogs were paraplegic, with a strong significance among DPN dogs (*χ*^2^(1, *N* = 361) = 25.562, *p* ≤ 0.001)—which is explained by the fact that DPN dogs are not able to respond to pain stimuli, reducing their reflexes and increasing the pressure on bone prominences, leading to recumbency ulcers [62,63]. Most of them occur in bone prominence regions due to the tissue ischemia caused by the pressure and blood circulation [64,65,66], and the two major sites in our study were the scapulohumeral joint and the femur greater trochanter region (Figure 4), in agreement with other authors [62,67,68,69,70].

Degenerative changes in muscles alter their contractile capacity [71,72], although in humans, it has been proven that exercises and general physical activity can restore neuromuscular function in the muscles [73,74]. These exercises were applied in this study, probably lowering the incidence of pressure sores in dogs with muscle atrophy, with no significance observed in either the paraplegic or tetraplegic dogs.

For the management of grades I and II pressure sores, trichotomy and cleaning with 0.05% chlorhexidine solution were performed, followed by application of therapeutic honey (antibacterial properties) or paraffine compresses. To increase the healing process, class IV laser therapy (Companion Therapy Laser ^®^, New Castle, DE, USA) was used (Figure 5), given the photobiomodulation effects in promoting cellular photochemical reactions. In this study, the contaminated wound protocol used was as follows: 2W, 30 s, each time at 20 Hz, 500 Hz, 5000 Hz and 10,000 Hz, in this order, with sessions for five consecutive days [14]. For grades III and IV pressure sores, surgical cleaning was required with debridement and approximation sutures, and then treated in the same way as mentioned above [62]. Patients with this clinical occurrence were maintained in soft beds, and positioning was mandatory every two hours and cotton donut bandages were placed [65,75]. For recurrent cases or with difficult healing, hyperbaric oxygen therapy was prescribed (HVM^®^, Boca Raton, FL, USA), accelerating the recovery process [76].

Additionally, the prevalence of respiratory problems was low, represented by aspiration pneumonia in 1.8% (9/488) and bronchopneumonia of other etiology in 0.2% (1/488) of dogs. This low percentage may be a consequence of hospital management with strict sternal recumbency rules [77,78], and night positioning every 4–6 h [75], depending on the risk and severity of pulmonary atelectasis, which is usually assessed by point-of-care ultrasound (POCUS).

Furthermore, the sternal position reduces the possibility of vomiting [79], mostly in brachycephalic breeds, thus reducing aspiration pneumonia, which was only observed in one brachycephalic dog (1/9). Of the 26% (127/488) patients with cervical lesions, associated with the neurological condition of tetraplegia, six dogs developed aspiration pneumonia, in agreement with studies that reported an incidence 18 times higher than in paraplegic dogs (*n* = 4 in this study) [77,80]. However, regarding the absence of sternal recumbency ability, there was a clear significance with aspiration pneumonia (*χ*^2^(1, *N* = 488) = 4.274, *p* = 0.039), with a prevalence of 88.9% (8/9).

As preventive measures, in the initial phase of hospitalization, the dogs were placed in passive station centers or physioballs (Figure 6) [11,81], and respiratory kinesiotherapy massages (e.g., coping) were performed, associated with nebulization in severe cases [75].

In addition, monitoring of urinary retention and emptying control was performed using POCUS (Figure 7A), allowing the evaluation of urinary sediment and, if necessary, when a UTI was suspected, a guide to perform cystocentesis [82]. In the gastrointestinal evaluation, with worsening of the clinical signs such as vomiting and diarrhea, POCUS was also performed to assess gastrointestinal mobility, based on the physiological number of four to five gastric-duodenal contractions per minute. The diagnosis of ileus was established when there was filling of the gastrointestinal lumen without visible contractions [83]. Still, the evaluation of pleural and lung space through POCUS (Figure 7B) was essential in all patients who remained in the lateral decubitus position, especially at night, and in whom clinical respiratory signs such as dyspnea appeared [84].

## 5. Conclusions

In this population of tetraplegic and paraplegic dogs undergoing FNR in a rehabilitation center, 79.5% showed clinical occurrences, highlighting the French Bulldog and the Dachshund as the most prevalent breeds. Thus, the highest prevalence of occurences was in the urinary system, with 58% of dogs having neurogenic bladder and 21.3% having urinary incontinence. The second most prevalent was the gastrointestinal system, with 35.5% of dogs having diarrhea and 20.5% having fecal incontinence. It is also important to mention that for the DPN dogs, 98.5% had at least one clinical occurence, and there were significant associations between them, fecal and urinary incontinence, and the thoracolumbar localization. In brachycephalic breeds, it was demonstrated that 97.3% of these dogs had gastrointestinal problems, 86% had ophthalmological disorders, and 74.1% had dermatitis.

In the present study, there was a low incidence of respiratory problems and UTIs, possibly due to the preventive measures carried out at the rehabilitation center, allowing to conclude that an early, synchronized, and methodical approach can help to reduce these percentages. The use of POCUS, either with the thoracic technique to avoid atelectasis or with the abdominal technique to monitor bladder and gastrointestinal mobility, associated with the early implementation of neurorehabilitation protocols such as positioning strategies, electrical stimulation, class IV laser therapy, exercises, and respiratory kinesiotherapy, may allow a better quality of life in patients and the reduction of euthanasia. 

## Figures and Tables

**Figure 1 animals-13-01164-f001:**
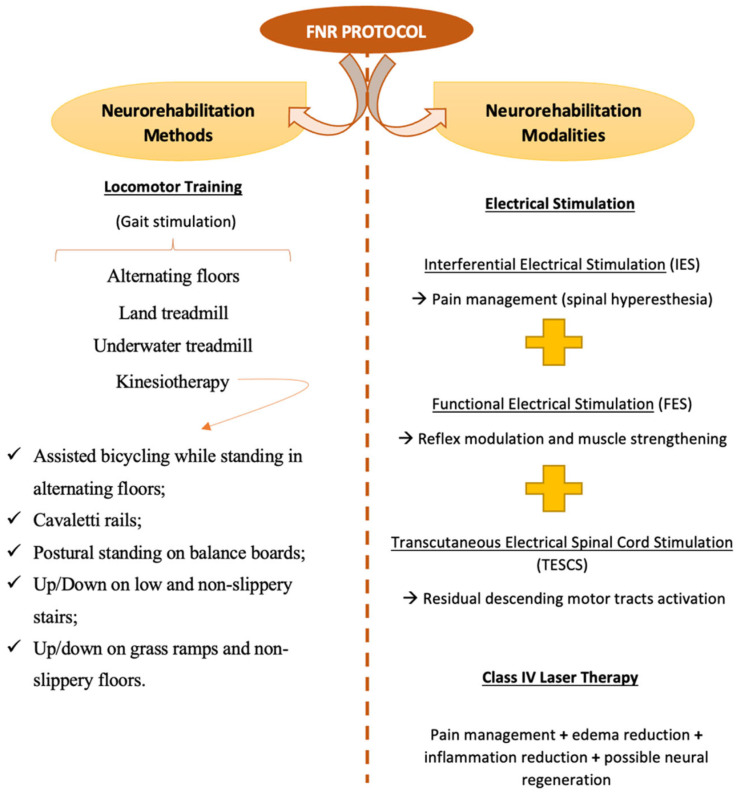
Overall FNR protocol prescribed for the study population (n = 488).

**Figure 2 animals-13-01164-f002:**
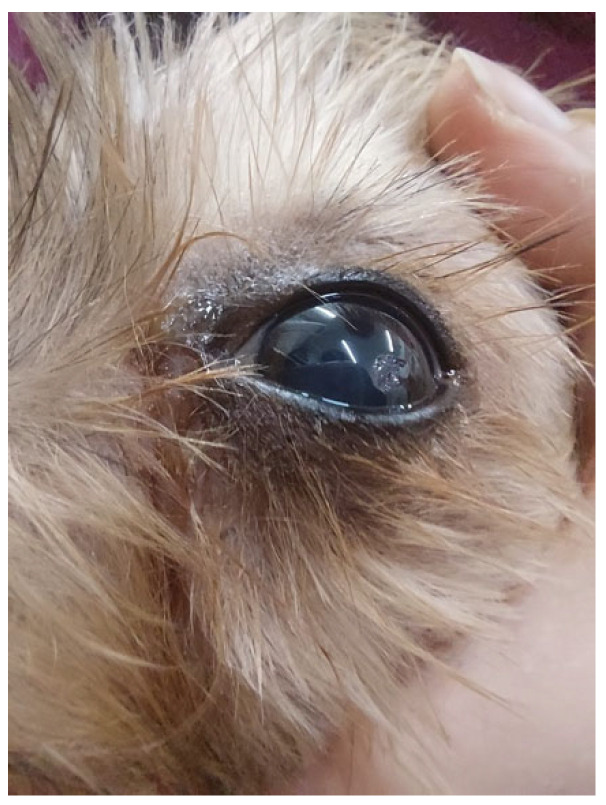
Corneal ulcer on the left eye of a dog.

**Figure 3 animals-13-01164-f003:**
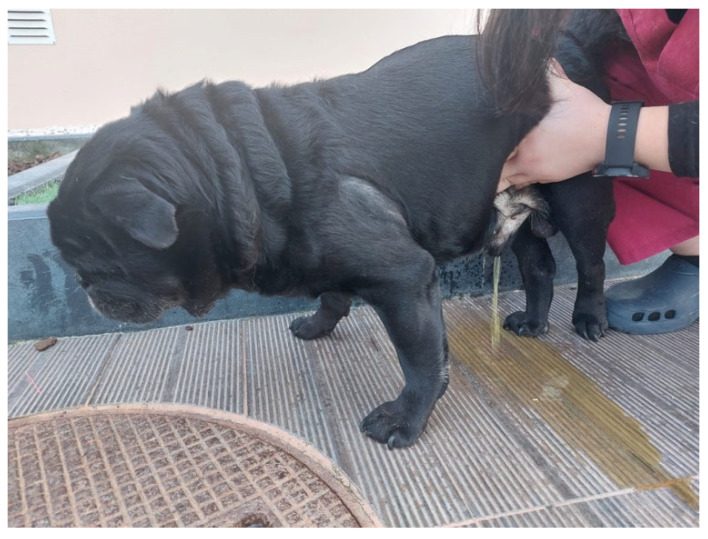
Manual bladder expression technique performed on a dog.

**Figure 4 animals-13-01164-f004:**
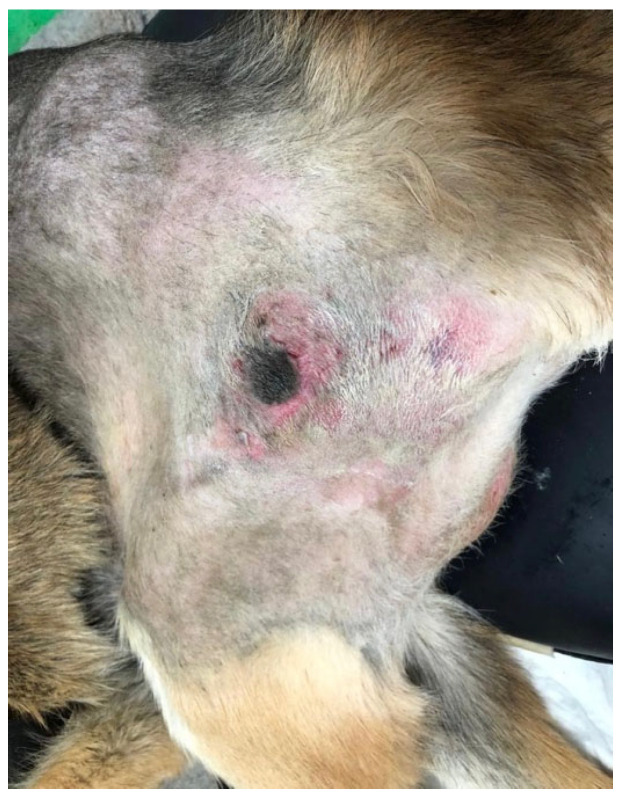
Pressure sore in the femur greater trochanter region of a dog.

**Figure 5 animals-13-01164-f005:**
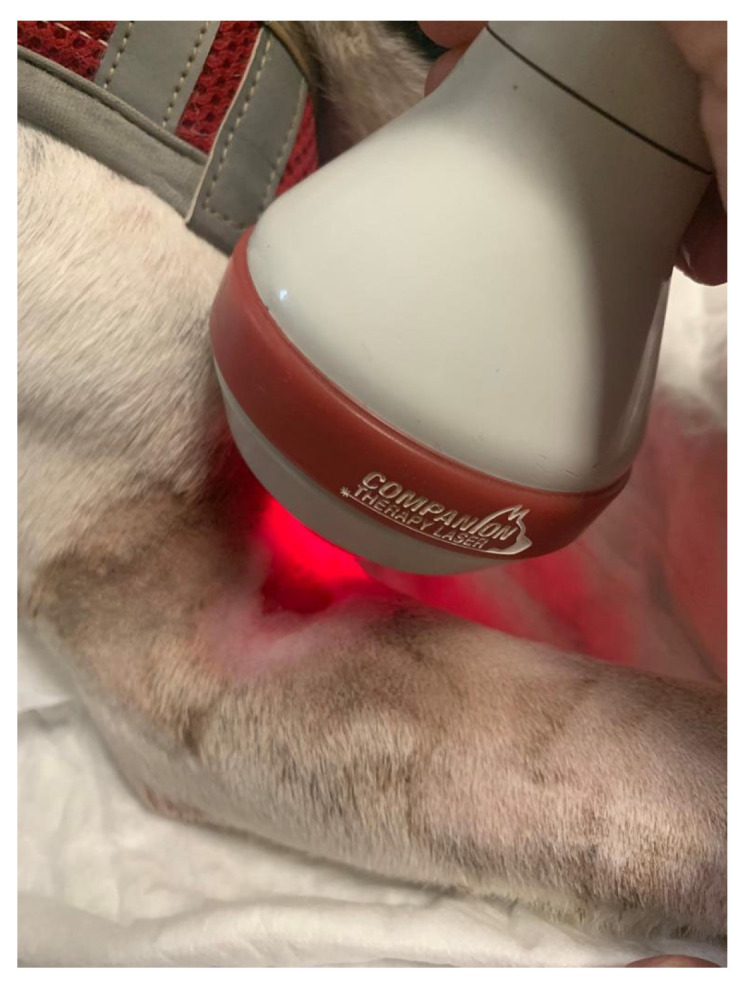
Class IV laser therapy performed on an elbow pressure sore of a dog.

**Figure 6 animals-13-01164-f006:**
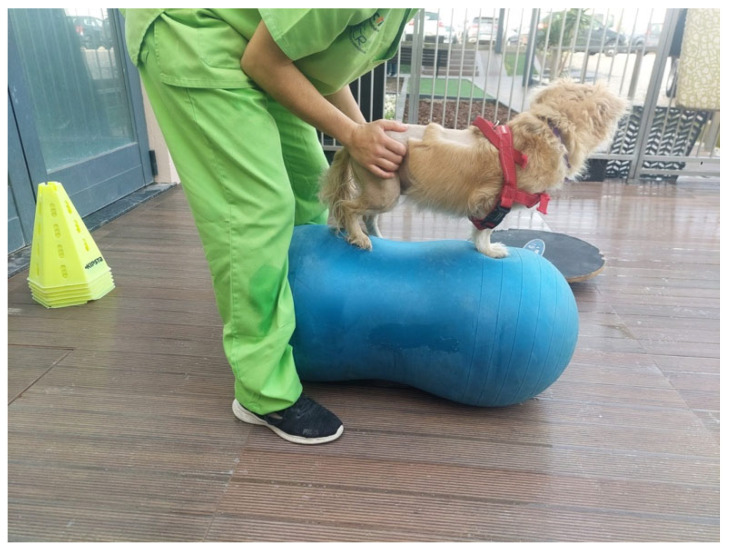
Assisted postural standing position of a dog on a physioball.

**Figure 7 animals-13-01164-f007:**
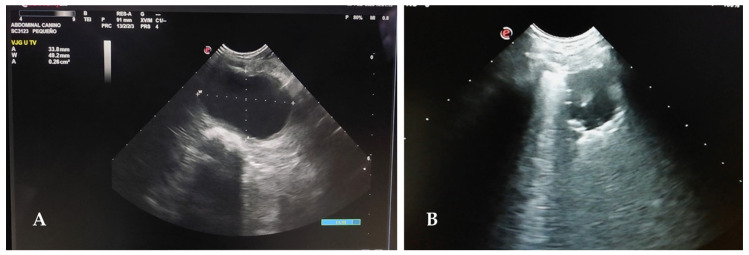
Veterinary point-of-care ultrasound (POCUS). (**A**) Abdominal POCUS screening of a bladder for urinary retention and sediment control. (**B**) Thoracic POCUS showing a consolidation in a dog with pulmonary atelectasis.

**Table 1 animals-13-01164-t001:** Sample characterization of the study population.

	Total (*n* = 488)
Age	<7 years old: 72.3% (353/488) ≥7 years old: 27.7% (135/488) Mean: 5.57 years old
Weight	<15 kg: 72.1% (352/488) ≥15 kg: 27.9% (136/488) Mean: 13.57 kg
Sex	Male: 59.8% (292/488) Female: 40.2% (196/488)
Breed	Pure-breed: 77% (376/488) Mixed-breed: 23% (112/488)
Neuroanatomical localization	Cervical: 26% (127/488) Thoracolumbar: 73% (355/488) Lumbosacral: 1% (6/488)
Etiology	Compressive: 93% (454/488) Non-compressive: 7% (34/488)
DP perception	DPN: 27.5% (134/488) DPP: 72.5% (354/488)
Sternal recumbency	Absent: 54.9% (268/488) Present: 45.1% (220/488)
Clinical Occurrences	Absent: 20.5% (100/488) Present: 79.5% (388/488)

DP, deep pain; DPN, deep pain negative; DPP, deep pain positive.

**Table 2 animals-13-01164-t002:** Clinical occurrences check list.

System	Occurrences	
Urinary	Neurogenic bladder	Yes □ No □
	Urinary incontinence	Yes □ No □
	Urinary infection	Yes □ No □
Gastrointestinal	Fecal incontinence	Yes □ No □
	Diarrhea	Yes □ No □
	Vomiting	Yes □ No □
Dermatological	Dermatitis	Yes □ No □
	Pressure sores	Yes □ No □
Respiratory	Kennel cough	Yes □ No □
	Aspiration pneumonia	Yes □ No □
	Bronchopneumonia	Yes □ No □
Musculoskeletal	Muscle atrophy	Yes □ No □
	Spasticity	Yes □ No □
Ophthalmic	Conjunctivitis	Yes □ No □
	Episcleritis	Yes □ No □
	Corneal ulcers	Yes □ No □
Others	Pyrexia	Yes □ No □
	Pancreatitis	Yes □ No □
	Discospondylitis	Yes □ No □
	Progressive myelomalacia	Yes □ No □

**Table 3 animals-13-01164-t003:** Descriptive analysis of age and weight (n = 488).

		Total (*n* = 488)
Age	Mean	5.57
	Median	5
	Mode	5
	Variance	9.046
	SD	3.008
	Minimum	1
	Maximum	16
	SEM	0.136
	Kolmogorov–Smirnov normality test	≤0.001
Weight	Mean	13.57
	Median	10
	Mode	7
	Variance	111.075
	SD	10.539
	Minimum	1
	Maximum	62
	SEM	0.477
	Kolmogorov–Smirnov normality test	≤0.001

SD, standard deviation; SEM, standard error of the mean.

**Table 4 animals-13-01164-t004:** Prevalence of clinical occurrences in the study population (*n* = 488).

System	Prevelance of Occurrences	
Urinary	Neurogenic bladder	58% (283/488)
	Urinary incontinence	21.3% (104/488)
	Urinary tract infection	6.1% (30/488)
Gastrointestinal	Fecal incontinence	20.5% (100/488)
	Diarrhea	35.5% (173/488)
	Vomiting	14.5% (71/488)
Dermatological	Dermatitis	22.1% (108/488)
	Pressure sores	19.5% (95/488)
Respiratory	Kennel cough	1.2% (6/488)
	Aspiration pneumonia	1.8% (9/488)
	Bronchopneumonia	0.2% (1/488)
Musculoskeletal	Muscle atrophy	31.1% (152/488)
	Spasticity	8% (39/488)
Ophthalmic	Conjunctivitis	16.6% (81/488)
	Episcleritis	10.2% (50/488)
	Corneal ulcers	2.9% (14/488)
Others	Pyrexia	5.1% (25/488)
	Pancreatitis	0.6% (3/488)
	Discospondylitis	0.2% (1/488)
	Progressive myelomalacia	4.5% (22/488)

## Data Availability

The data presented in this study are available upon request from the corresponding author.

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
