# Peer review of "Clinical Occurrences in the Neurorehabilitation of Dogs with Severe Spinal Cord Injury"

_animals, 2023, doi:10.3390/ani13071164_

Round 1
Reviewer 1 Report
The manuscript presents the results of a prospective observational clinical study in a population of tetraplegic and paraplegic dogs, with a considerable size sample (n=488). The prevalence of clinical occurrences in this kind of patients while hospitalized in a rehabilitation center is presented and discussed.
This work is very interesting and of high clinical relevance, since risk predisposing factors, preventive measures, and therapeutic options were analyzed and discussed. It also addresses the subject from a One Health approach, with frequent reference to human patients with analogue affections.
The cited references are relevant for introducing the addressed issues and discussing the results.
The statistical analysis methodology is sound.
The formatting of the different sections should be revised (paragraph and line spacing).
The manuscript lacks the Simple summary.
The abstract would benefit of revision, namely for clarity.
The quality of the manuscript is affected by lack of clarity in many sentences and several grammar and phrase construction problems. The manuscript should be subjected to extended English language revision. A few examples of needed corrections (but not limited to):
Line 16 to 18
The highest prevalence of observed occurrences affected the urinary system (58% of neurogenic bladder and 21.3% of urinary incontinence), followed by the gastrointestinal system (35.5% of diarrhea and 20.5% of fecal incontinence). A low incidence of respiratory problems (e.g., pneumonia) and urinary tract infections was observed"
Line 43
Most of the time, these patients may have reduced or absent anorectal sensitivity and voluntary control of the external anal sphincter may be compromised, leading to a high risk of fecal incontinence, which may also increase secondary to intensive neurorehabilitation training…
Line 68 “neurological dogs” replace by dogs with neurological conditions; the same for identical situations (such as “cervical dogs”)
Line 100 “The examination took place in a controlled environment”; the preposition “on” is frequently misused throughout the document.
Line 108 “each of the hind and forelimbs”.
Line 78 “there was 79.5% (388/488) dogs” replace by “At least one clinical occurrence was observed in 79,5% (388/488) of the dogs included in the study, while 20,5% (100/488) presented none.”
Table 2 Discospondylitis, not discoespondylitis
Line 200 “cervical patients,” consider replacing by "patients with cervical lesions" and re-organizing the text to include this in the paragraph associating the location of the lesions to the most common occurrences.
Reviewer 2 Report
First of all, I must express that the article under review deals with a subject that has been little addressed and interesting; however, in my opinion the authors make too many comparisons (that are not related to neurological pathology) and introduce variables of clinical findings, which I believe are casual, so I do not think it is of clinical interest.
For example, in the case of brachycephalic breeds, it seems to me that the reported clinical occurrences are related to the idiosyncrasy of the breed and not to the pathology.
Regarding the common findings that are observed due to neurological disease, they do not provide novel data of interest.
For all these reasons, I am sorry to have to decline its publication in the magazine.
Reviewer 3 Report
This is a very interesting article from a clinical point of view.
The collection of at least 7 years of clinical data from 488 dogs provides a great help to veterinary rehabilitators to know where to focus their treatment during neurorehabilitation.
Although it may seem simple from an experimental point of view, it nevertheless describes a large body of work that will be of great interest.
Attached are some minor revisions and questions that I have had about the research.
.-.-.-.-.-.-.-.-.-.-.
3.-It would be advisable to include Severe spinal cord injury in the title (instead of the abbreviation). To facilitate the search for clinical veterinarians, especially taking into account that SCI does not appear in the keywords. Perhaps, to make it more accessible, it would be advisable to include Severe spinal cord injury (not the abbreviation) in the title and, in the keywords, to include SCI.
18.-Delete a "re".
77-81. Have available the models of the report consented by the owners as well as the document of approval by the ethics committee of the year of the start of the study (2016).
96.-Maybe "Legend" can be deleted. In the same table (1), in the Neuroanatomical localization row, between Cervical, Thoracolumbar and Lumbosacral, it does not reach 100%. Perhaps it would be advisable to revise the tenths so that, among all the locations, they reach 100.
101.- Canon camera model (country of manufacture missing?).
Table 3.-Variance 111.075 kg. I imagine you have checked it. Is it correct?
160.- A "comma" is missing between Bullmastiff (n=2),; English Setter (n=2).
Point 4.-Discussion. Revise line spacing format.
259.-As for weigh, an aerage of 13.57 kg (change to a point).
294.-(...) for at least 15 days to reach ambulation of 68% [17,32-35]. Change the "comma" to a "full stop" (17.32).
343-345. How many dogs had to be euthanised, and is there any data from your work on this?
394.-Revise the commercial name of the laser equipment (place of manufacture?) in order to be able to replicate your study.
401-403.-On how many patients was hyperbaric therapy used? with hyperbaric chamber?
Round 2
Reviewer 2 Report
As I already commented after correcting the first version of the article, it does not seem to me that its content is publishable, for the same reasons that I already gave above. Therefore, in my opinion it should not be considered for publication.